# Impact of the Tumor Microenvironment for Esophageal Tumor Development—An Opportunity for Prevention?

**DOI:** 10.3390/cancers14092246

**Published:** 2022-04-30

**Authors:** Martin Borgmann, Michael Quante

**Affiliations:** Department of Internal Medicine II, University Medical Center Freiburg, 79106 Freiburg, Germany; martin.borgmann@uniklinik-freiburg.de

**Keywords:** esophageal adenocarcinoma, prevention, carcinogenesis, tumor microenvironment, Barret’s esophagus

## Abstract

**Simple Summary:**

Researchers increasingly appreciate the tumor microenvironment (TME) for its role in the development and therapy resistance of cancers like esophageal adenocarcinoma. A better understanding of the TME fueling carcinogenesis is necessary for tailored prevention and therapies. Here, we highlight recent insights into tumor initiation, interactions with the immune system and possible novel preventative measures.

**Abstract:**

Despite therapeutical advancements, and in contrast to other malignancies, esophageal adenocarcinoma (EAC) prognosis remains dismal while the incidence has markedly increased worldwide over the past decades. EAC is a malignancy of the distal esophageal squamous epithelium at the squamocolumnar junction with gastric cells expanding into the esophagus. Most EAC patients have a history of Barret’s esophagus (BE), a metaplastic adaption to chronic reflux, initially causing an inflammatory microenvironment. Thus, the immune system is highly involved early on in disease development and progression. Normally, anti-tumor immunity could prevent carcinogenesis but in rare cases BE still progresses over a dysplastic intermediate state to EAC. The inflammatory milieu during the initial esophagitis phase changes to a tolerogenic immune environment in BE, and back to pro-inflammatory conditions in dysplasia and finally to an immune-suppressive tumor microenvironment in EAC. Consequently, there is a huge interest in understanding the underpinnings that lead to the inflammation driven stepwise progression of the disease. Since knowledge about the constellations of the various involved cells and signaling molecules is currently fragmentary, a comprehensive description of these changes is needed, allowing better preventative measures, diagnosis, and novel therapeutic targets.

## 1. Introduction

Most of the research and clinical focus in gastrointestinal cancer has been on cell autonomous mechanisms in the epithelial compartment. However, accumulating evidence demonstrated that epithelial cells, and specifically stem cells, are strongly influenced by the host microenvironment [1]. Cancers originating in the context of chronic inflammation are likely driven by environmental mediators and cells, which together establish an aberrant tumor microenvironment that predisposes to cancer initiation and promotes tumor progression [2]. Moreover, it has been suggested that the lifetime risk of cancers is strongly correlated with the total number of stem cell divisions, suggesting that only a third of the variation in cancer risk is attributable to external factors or inherited predispositions and therefore likely more to the host microenvironment [3].

In Esophageal Adenocarcinoma (EAC), the tumor progressively evolves in an inflammatory process from the precursor lesion Barrett’s Esophagus (BE), which is primarily caused by chronic reflux (gastroesophageal reflux disease, GERD). This disease is frequent in obese people which may explain why EAC is most common in industrialized countries [4]. The stepwise progression from BE over dysplasia to EAC is observed in roughly 0.1–0.3% of BE patients per year, for which tobacco smoking was identified as a risk factor [5,6]. BE and EAC have been associated with similar risk factors, including GERD, Caucasian race, male sex, increasing age, and obesity. The prevalence of BE has also increased greatly over the last decades, resulting in many individuals “at risk” for this fatal malignancy. However, most EAC patients were not previously diagnosed with BE, suggesting that incidences are greatly underreported [7]. EAC poses a major global health burden because it has a relatively high mortality rate of ≥80%, ranking as the 6th deadliest malignancy in 2015 [8,9]. Despite the inflammation during development, cancer immunotherapies show a low response rate [10]. To design better preventive and therapeutic strategies for BE and EAC, it will be of immense importance to understand the function and contribution of the tumor microenvironment (TME). Recent research increasingly focuses on the characterization of the TME, and this review aims to summarize the current knowledge thereof in EAC with a focus on immunologic aspects.

## 2. Pathophysiology and Carcinogenesis of EAC

The incidence of gastroesophageal junction cancer, comprising both esophageal (EAC) and junctional gastric adenocarcinomas, has increased dramatically in Western countries with a 5-year overall survival rate of 15%. The rapid increase is associated with obesity and gastro-esophageal reflux disease (GERD), which are thought to promote Barrett’s esophagus (BE). Better understanding of the pathogenesis of this lethal disease would allow improved cancer prevention, early (endoscopic) detection and therapeutic options. BE begins at the very distal esophagus, contiguous with the gastric cardia and is described as an archetypal metaplastic condition comprising a mosaic of gastric and intestinal cell types. Metaplastic changes of the squamous epithelium to a columnar phenotype are followed by a stepwise progression from non-dysplasia (NDBE) over low-grade dysplasia (LGD), high-grade dysplasia (HGD) and the carcinoma. The development of EAC is associated with inflammation causing an infiltration by immune cells. For a long time it has been assumed that such metaplasia was originating from squamous epithelial cells and was associated with an increased risk of malignancy.

In general, metaplasia is defined as replacement of differentiated cells with other mature differentiated cells that are not normally present in a specific tissue. This is distinct from trans-differentiation, a process in which a differentiated cell type converts into a completely different cell type present in the tissue. This has long been searched for but not detected in esophagus tissue so far. Metaplasia may be induced by some sort of abnormal stimulus (i.e., acid, bile acid, inflammation, cigarette smoke, and alcohol), and with persistent exposure is thought to progress to dysplasia and occasionally EAC. Given that the native esophagus is squamous, the glandular phenotype of EAC is likely inextricably linked to BE metaplasia.

New data supporting a gastric origin of EAC/BE have emerged in recent years from both deep analysis of human samples and experimental results from human derived cells and mouse models. The hypothesis that BE originates in the gastric cardia was proposed in 2012, based on findings in lineage tracing studies in a BE (L2-IL-1β) mouse model, which recapitulates the histologic progression from esophagitis to dysplasia [11]. Lineage tracing allows for the genetic definition and tracking of stem cells and their progeny and can help determine the cellular origin of neoplasms. These experiments point to cells such as Lgr5^+^ intestinal stem cells (ISC) from the cardia as the origin of BE [12]. The L2-IL-1β mouse model overexpresses IL-1β in the upper gastrointestinal tract [11]. This pleiotropic cytokine is an upstream mediator for IL-6 and TNFα in GERD-associated esophagitis, which in turn drive disease progression [13]. This leads to the histologic progression from esophagitis to dysplasia, similar to human histopathology, allowing the tracing of stem cells and their progeny in BE. Subsequently, in 2017 The Cancer Genome Atlas Research Network (TCGA) demonstrated in comprehensive molecular genomic profiling of both esophageal and gastric cancers (GC), the distinct features of the two histological subtypes of esophageal cancer, EAC and esophageal squamous cell carcinoma (ESCC). In contrast to EAC, ESCC shows much greater similarity to head and neck SCC. Furthermore, joint analysis of EAC and GC could not identify features clearly distinguishing these two cancers, suggesting a shared origin [14]. Genetic results from TCGA are consistent with recent epigenetic studies of BE relative to normal gastric and esophageal tissues, which also demonstrated evidence for a gastric origin of BE [15]. Another recent study utilized comprehensive single-cell transcriptomic profiling, in silico lineage tracing, and mutation analyses from human tissues spanning the proximal stomach to squamous esophagus from healthy and diseased donors. The results showed that BE originates from gastric cardia progenitors through distinct transcriptional programs [16]. This study also experimentally determined that organoid cultures of human gastric tissue have the capacity to differentiate into BE. The emerging view of BE/EAC as originating from gastric tissue is consistent with key pathologic findings that BE always begins at the very distal esophagus, contiguous with the gastric cardia, and that BE comprises a mosaic of gastric and intestinal cell types. Moreover, this metaplasia is largely indistinguishable from intestinal metaplasia in the stomach. This new thinking echoes the original descriptions of metaplasia of the distal esophagus by Norman Barrett, who assumed at the time that BE resulted from proximal migration of stomach epithelium [17,18].

The metaplasia at the GEJ involves a deviant differentiation of the ISC. The reason for this is probably the deficient healing of the squamous epithelium as columnar epithelial progenitors are more resistant to acid/bile injury. Like later stages in the development of EAC, BE displays a relatively high mutational rate. Clonal complexity is already detectable before the malignancy or even metaplasia is observable [19]. This demonstrates an ongoing evolutionary process which is fueled by chronic inflammation caused by GERD. During reflux, contents of the duodenum can get to the GEJ. Here, bile acids such as deoxycholate can cause DNA damage [11]. The effect might be aggravated by a high-fructose-diet, common in obese people, because it reduces the number of goblet cells which produce protective mucus [20]. Through the genomic evolution, GERD is driving a selection process for clones with cancer-associated mutations, as described by Vogelstein in his adenoma–carcinoma sequence model [21,22,23]. Ultimately, this increases the risk for a tumor formation, possibly without metaplastic development.

The histological diversity in BE and EAC is accompanied by a complex mutational landscape with clonality within BE segments but genomic instability in EAC [24,25]. In fact, chromosomal instability has been demonstrated in dysplastic BE patients while hardly any instability was detectable in non-dysplastic patients based on single-cell DNA sequencing [26]. Similarly, other genomic alternations such as copy-number alterations or loss of heterozygosity are only found in patients with progression to EAC [24]. Genetic mutations frequently found in EAC patients include the tumor suppressors p16 (CDKN2A) or p53 (TP53) and the oncogene NOTCH1 [27,28]. Upregulated genes in EAC include the proto-oncogene YAP1 of the Hippo pathway as well as one of its targets SOX9 [29]. In a study of invasive gastric adenocarcinoma, genes associated with cancer stem cell properties (ALDH3A1, SOX9, EGR3 and HES-1) were upregulated too [30]. Given that this type of cancer shares etiology, a similar upregulation can be expected in EAC.

Accumulating evidence indicates that BE and EAC pathogenesis involves the aberrant differentiation of stem or progenitor cells at the squamocolumnar junction (SCJ). Again, the mouse model suggests that a combination of risk factors such as high fat diet, GERD, bile acid, microbiome or host driven distinct inflammation control stem cell homeostasis at the SCJ. This can lead to malignant transformation and tumor growth in combination with characteristic genetic alterations. The high mutation rate and clonal complexity of BE is evidence of the ongoing evolutionary process that begins long before the development of a detectable malignancy or even metaplasia. A stem cell niche, within the BE segment but also in the relative stomach, represents a clonal mosaic, where genetically distinct (stem cell) clones compete, leading to a dynamic equilibrium of subclone expansion and retraction. The inflammatory reflux-induced microenvironment fuels evolution, selecting for clones harboring cancer-associated mutations and increasing the chance of cancer development with or without metaplastic development. Environmentally driven inflammation is likely affecting stem cell proliferation or differentiation and should be considered as the most important risk factor. This calls the well-established theory of a metaplasia to dysplasia sequence of gastric or esophageal tumor development into question with significant impact on other tumor diseases (i.e., colon, breast, pancreas) and cancer prevention. Metaplasia itself might thus not be a typical precursor lesion. Given the stable nature of BE, carcinogenesis probably reflects the expansion of an altered stem cell population due to changes in stromal niche factors.

In the clinic, risk prediction models that consider such microenvironment and epidemiological risk factors will need more implementation and validation. BE patients are frequently enrolled into surveillance programs aiming to detect dysplasia or EAC at an early stage to improve patient outcomes. When performed correctly, such programs should result in increased dysplasia detection and reduced EAC mortality, yet most BE patients undergo long-term invasive surveillance efforts to avoid EAC that will never arise [31]. Thus, as a malignant precursor, BE metaplasia may only represent the tip of the iceberg, and inflammation at the GEJ may be the underwater bulk that has not been sufficiently recognized to combat cancer development.

## 3. The Tumor Microenvironment Fuels Esophageal Carcinogenesis

With the emerging field of cancer immunotherapy it has become clear that a better understanding of the tumor microenvironment (TME) is necessary to understand cancer progression and therapy resistance [32]. Growing evidence from the mouse model points to an important role of the microenvironment in triggering many of the earliest events of tumor initiation. Stem cells likely reside in a niche that maintains them in a stem-like state, and tumor-initiating cells require a dedicated microenvironment to control self-renewal and maintenance of an undifferentiated state. In esophageal carcinogenesis, chronic inflammation promotes the proliferation and survival of malignant cells by subverting innate and adaptive immune responses

### 3.1. The Mesenchymal Contribution

The TME comprises the extracellular matrix (ECM) and a cellular compartment. The ECM in a tissue influences the interactions of the stromal cells with the TME and the immune system which is important for anti-tumor immunity. It consists of many adhesion molecules and a scaffolding structure. In addition, fibroblasts are a major stromal cell type that provides structure molecules for the ECM and produces cytokines [10].

Apart from cancer cells and infiltrating immune cells, the stromal microenvironment of tumors includes a mixture of mesenchymal cells, comprised mostly of cancer-associated fibroblasts (CAFs). CAFs have been linked to poor prognosis in solid malignancies and resistance of EAC to immunotherapy [10]. Activated fibroblasts contribute to tumorigenesis by enhancing proliferation and tumor-initiating capacities, and by recruiting and polarizing cells of the adaptive and innate immune system towards a tumor-promoting phenotype [33]. They express vimentin and α-smooth muscle actin (α-SMA), resembling normal myofibroblasts that are present in the gastrointestinal mucosa and develop partially from bone marrow mesenchymal stem cells. Here, they contribute to the physiologic BM niche and mesenchymal stem cell self-renewal [34,35]. They are recruited to the tumor site via TGF-β and CXCR4/CXCL12 signaling [34]. Other reported CAF origins include local fibroblast and carcinoma cells, which underwent mesenchymal transition [36].

It is well known that in solid cancers like EAC, CAFs substantially contribute to tumorigenesis and metastasis [10,34]. Their tumor-supportive characteristic stems from a multitude of actions, for instance by remodeling the ECM. The expression of lysyl oxidase (LOX) family enzymes results in a crosslinking and thereby maturation of the collagenous structure [37]. In addition, CAFs can excessively produce collagen while altering the compositions of the present types of collagen [38]. Together, these mechanisms are thought to make the ECM stiffer and harder for potentially cytotoxic immune cells to penetrate, as these express limited amounts of proteolytic enzymes such as matrix metalloproteinases (MMP) [39]. In comparison, malignant tumors do express MMPs, which is crucial for their invasiveness [39,40]. Interestingly, it was shown that CAFs can travel alongside cancer cells via the bloodstream and help with extravasation at metastatic sites [41]. CAF-originating MMPs also pose a pro-angiogenic factor by creating favorable conditions for angiogenesis through the degradation and remodeling of the ECM, as reviewed by Wand, Zhang and Fan [42]. The authors give more examples for factors from CAFs that support angiogenesis, like the expression of vascular endothelial growth factor (VEGF) or stromal cell-derived factor 1 (SDF-1), which can recruit endothelial progenitor cells via CXCR4.

### 3.2. The Ignition of Immune Reactions

Inflammation as a hallmark of cancer [43] is known to play an essential role in carcinogenesis and progression of most cancer types [44]. Chronic irritation in the distal esophagus through acid and bile reflux causes inflammation, leading to the recruitment of pro-inflammatory immune cells, and to the stimulation of epithelial cell proliferation, survival, and migration, as depicted in Figure 1 [45,46]. Immune cells secreting signaling molecules to promote tissue healing infiltrate the tissue as well [47]. Several studies have shown that IL-1β, present in esophagitis, induces IL-6 production, driving inflammation [10,48].

In the L2-IL-1β mouse model, IL-8 promotes development of Barrett’s esophagus and esophageal adenocarcinoma in part through recruitment of immature myeloid cells [49]. A study by Münch et al. using the model showed an influx of Lgr5^+^ CXCR2^+^ cells in dysplastic tissues, indicating that IL-8 might also serve as a chemoattractant for the ISC [49]. Survival and proliferation of ISC are regulated by the stem cell niche, like neighboring differentiated epithelial cells or pericryptal myofibroblasts [50]. Major signaling pathways include the Wnt and Notch, as well as Bone Morphogenetic Proteins (BMP) and Hedgehog pathways [51]. Of those, Notch and Wnt have already been identified to be involved in the BE to EAC sequence [52,53]. Moreover, in several cancers, including EAC, IL-6 induces epithelial-to-mesenchymal transition (EMT) by upregulating cancer stem cell associated genes [10,54]. EMT is a key event in metastasis and increases therapy resistance [55,56].

Cancers that develop after chronic inflammation like EAC are usually substantially driven by infiltrating immune cells. This has led scientists to endeavor the establishment of a quantifiable measurement tool, the “immune score”, first described for colorectal cancer [57]. The score includes the infiltration of cytotoxic CD8^+^ T cells (CTLs) and memory T cells (T_cm_) [58]. It has a relatively high prognostic value, possibly superior to the classical TNM staging, and is therefore interesting as an adjunct factor considering adjacent immunotherapies [59]. Indeed, CTLs and the immune system as a whole play a crucial role in averting cancer, hence higher scores represent a better prognosis. The immune system has the potential to recognize mutations as “altered self”. For example, Segal et al. performed an in silico analysis, estimating about 10 or 7 new and unique MHC-binding peptides per HLA allele in breast and colorectal cancer, respectively [60]. However, despite the constant vigilance of the immune system, tumors can thrive by evading the antitumor immune response. The TME of both malignant and non-malignant tumors is comprised of a network of cells that have adapted an immunosuppressive phenotype.

Although the immune cell infiltration in tumors is heterogenous between patients, similarities can be found within different types of cancers [32]. Figure 2 gives an overview of the interacting immune cells in EAC: dendritic cells (DCs), tumor-associated macrophages (TAM), CTLs, T-regulatory cells (T_reg_), natural killer (NK) cells, B cells, and myeloid derived suppressor cells (MDSCs, or immature myeloid cells, iMCs) and mast cells (MCs) [32,46,59,61,62]. Lagisetty et al. found that eosinophils, a common cell type in the esophagus, vanish during BE to EAC progression accompanied by an increase of the immune suppressive T helper subset Th_2_ and a drop in CD8^+^ T cell population after progression from high grade dysplasia to EAC [61]. In their study they also show a shift from inflammatory immune cell markers to upregulated inhibitory markers like the PD1/PDL1-axis, based on RNA sequencing. This correlates with the clinical observation, that patients with eosinophilic esophagitis never develop any tumor disease of the esophagus, a correlation that needs to be evaluated in future studies [63]. In contrast, other inflammatory conditions such as lichen planus may lead to squamous cancer [64].

### 3.3. Dendritic Cells

Dendritic cells (DCs) can process tumor-derived antigens and activate antigen-specific CD8^+^ naïve T cells in the tumor-draining lymph node (TDLN) [65]. In the context of EAC, only a few studies suggest a role for DCs. DCs have access to large amounts of tumor antigens, including soluble mediators such as endogenous danger signals from necrotic cancer cells (DNA, HMGB1, S100), that are able to activate DCs [66]. Since the TDLN is the site to which DCs originating from the tumor migrate and neo-antigens from the tumor drain, the TDLN is crucial for the outcome in the development of cancer as well as effective immunotherapy [67]. The TDLN, however, is notably more tolerogenic compared to other LNs from the same animal [67]. DCs isolated from primary tumors as well as TDLNs are phenotypically immature and can poorly stimulate T cells. Immature DCs can inhibit effector T cell responses by inducing anergy, a defense mechanism that evolved to prevent autoimmunity [46,67]. DCs readily mature in vitro by pro-inflammatory stimulation but not in vivo, underlining that there is more to this phenomenon than just immaturity.

Bobryshev et al. reported that DC numbers in human biopsies increase in EAC compared to BE, but their mechanistic role in the TME for EAC development remains unresolved [62]. The authors discussed that increased numbers of DCs in several cancers appear to have lower allostimulatory activities compared to peripheral DCs. Their contribution may also be simply due to the defective maturation, which indirectly supports immune escape [68]. It has been shown that DCs are required for a proper response to treatment targeting the PD-1 axis [69]. Higher expression levels of PD-L1 by cancer cells in EAC significantly increases patient disease free survival after immune checkpoint inhibitor therapy [70].

### 3.4. T Cells

CD4+ T helper cells can be divided into several subsets, including Th_1_, Th_2_, Th_17_ and regulatory T cells (T_reg_s). Of those, T_reg_s have become a prominent member involved in cancer progression. They are induced by IL-2 and TGF-β and are necessary for peripheral tolerance [71]. Their normal effector function is to balance immune reactions, mainly by secreting IL-10 and TGF-β, which is especially important for the homeostasis of gut mucosa. Many cancers exploit these cells for immune evasion, exacerbating prognosis of most cancers. Surprisingly, in cancers of the GI tract their presence seems to be favorable, e.g., colorectal cancer [72,73] and EAC [74]. In CRC this observation could be explained by the high translocation of bacteria, which is normally accompanied by Th_17_ cell responses. These can have pro-tumorigenic effects that are inhibited by the T_reg_s. The direct role of microbiome alterations in the context of EAC is currently elusive but the presence of Th_17_ cells is assumed. Given the possible protective role, there is an urgent need to better understand the role of T_reg_s in EAC. In CRC, for instance, adoptively transferred T_reg_s suppresses inflammation-induced tumorigenesis, likely by inhibiting the formation of the inflammatory network [75,76,77,78]. This implies a beneficial role for the prevention of the BE to the EAC sequence.

Th_17_ cells differentiate from naïve T cells as well, in response to TGF-β, IL-6 and IL-23, which are all present in the TME [79,80]. They are characterized by the secretion of IL-17 and are associated with poor prognosis [72,79]. While not fully proven yet, the presence of Th_17_ cells in EAC is strongly implied by high levels of IL-17 in the TME [81]. Moreover, Th_17_ cells from other cancers constitutively express CCR4 and CCR6, and their corresponding chemoattractants CCL22 and CCL20 were found in the TME of esophageal cancers patients [80]. However, in the respective study, ESCC and EAC patients were mixed in the analysis.

The T cell subsets Th_1_ and Th_2_ are at play in the progression from BE to EAC too. Under physiological conditions, Th_1_ mediates tumor rejection by producing, e.g., TNFα and IFNγ and eliciting cell-mediated killing [82]. On the contrary, Th_2_ associated cytokines such as IL-4, IL-6 and IL-10 support tumor growth and suppress cellular immunity [83]. The balance between these two cell types shifts during carcinogenesis. In the initiating steps during esophagitis, squamous epithelial cells secret IL-1β and IL-8, as evident by both mouse and human data [49,84]. This pro-inflammatory environment changes after the development of BE, when patients have increased levels of mostly IL-10 and IL-4, suggesting a Th_2_-like response [84]. This milieu also supports the metaplastic changes in BE through the induction of the MUC2 gene by IL-4 [85]. Importantly, this state may support tumor development through the suppression of cell-mediated anti-tumor response [83]. In dysplasia, the balance shifts yet again back to Th_1_ with increase in IFNγ IL-1β, IL-2 and IL-8 levels [86,87].

### 3.5. MDSCs

MDSCs are pathologically activated immature myeloid cells (iMC) that are defined as CD11b^+^Gr-1^+^(Ly6G-Ly6C^high^) [88]. In cancer-free animals, they are detectable as iMCs in the bone marrow and occasionally the spleen, but accumulate in large numbers in the spleen under chronic inflammation and in cancer, where they become tolerogenic MDSCs [88,89,90]. MDSCs can suppress T cell functions in multiple ways, e.g., by transforming them into T_reg_s, expanding pre-existing T_reg_s or depleting the T cell essential amino acid arginine [91,92]. Besides this, they can also directly promote tumor progression and metastasis in an invasive colon cancer mouse model [93,94,95]. Cis-Apc/Smad-4 tumorigenesis is characterized by high iMCs infiltration at the invasion fronts. Here, they produce MMP2 and MMP9, which can degrade collagen IV, the predominant type in the basal membrane. Multiple studies have pointed out that stromal and tumor cell-derived IL-6 plays an essential role in the generation and activation of MDSCs for SCCs [96,97]. In a transgenic IL-8 mouse model, it was previously shown that this cytokine accelerated gastric and colon cancers through the recruitment of MDSCs [98].

### 3.6. TAMs

The polarization state of macrophages within the TME is divers and ranges from the rather tumoricidal M1-like, classically activated macrophages to the tumorigenic alternatively activated, M2-like phenotype. A clear distinction in vivo can be challenging due to sometimes overlapping cytokine profiles, as opposed to the clear polarization that can be induced in vitro [99]. In spite of the discrimination between M1 and M2 being oversimplified and perhaps outdated, TAMs are alternatively activated macrophages with tissue growth supporting and immune-suppressing functions, characterized by high levels of IL-10 production [100]. Within the M2-spectrum, they belong to the M2d subset, which is induced by IL6 [101]. Since IL-6 is a relevant cytokine in EAC, it could be assumed that this is one of the mechanisms that induces TAMs here. Other generally described mechanisms are Il-4 and hypoxia [102]. Blood monocytes are recruited to the tumor site via IL-8 where they differentiate to TAMs, but TAMs can also originate from mesenchymal-derived tissue resident macrophages [99]. TAMs express, e.g., angiogenic VEGF, IL-10, IL-12, TNFα and TGF-β [103]. Hence, their presence in cancers is associated with poor survival, which is also true in EAC [101]. Cao et al. also reported that the M2/M1 ratio was higher in late stage and metastatic patients [101]. To analyze the induction of TAM, they cocultured an EAC cell line (SKGT) with a macrophage cell line (THP1), which resulted in M2d polarization.

### 3.7. Neutrophils

Analogously to the plastic polarization of macrophages (M1/M2), neutrophils can also undergo a transformation from anti-tumor to a tumor-supporting phenotype (N1 to N2) [99]. Physiologically this aids in tissue regeneration after resolving an infection because of the collateral damage inflicted by neutrophilic activity. The pro-tumorigenic phenotype is mainly induced by TGF-β but it is uncertain whether these tumor-associated neutrophils (TAN) actually display a polarized phenotype or just a varying degree of activation [99]. Moreover, it remains open whether neutrophils have the same plasticity as T cells or macrophages and if the N2-like phenotype is reversible [104]. Our study implicates TANs in playing a role in EAC in IL-1β-overexpressing mice (L2-IL-1β mouse model) [49]. In this model, mice develop Barrett’s esophagus with similar histopathology to humans, as described above. Dysplastic tissues from mice fed with a high-fat diet produced elevated levels of CXCL1, the murine functional homolog of CXCL8. Accordingly, these mice had increased numbers of neutrophils in the tissue. Moreover, NK cell populations were reduced, suggesting that neutrophils inhibit NK cell degranulation, but the mechanism was not elucidated [49,105]. In alignment with this, another study found an increasing IL-8 expression during progression as well as higher neutrophil numbers in patient samples based on bulk RNA sequencing from tissues of different stages of the BE to EAC sequence [61].

### 3.8. NK Cells

NK cells belong to the lymphoid lineage but are generally considered as innate immune cells due to their lack of gene rearrangement. One of their major tasks is to detect and eliminate aberrant cells, e.g., cancer cells. They potentially directly recognize cancer cells, as these often downregulate MHC-1 expression as an immune evasion strategy (missing-self hypothesis) or indirectly via their Fc gamma receptor (antibody-dependent cellular cytotoxicity, ADCC) [106]. In response to activation, they secrete IFNγ, which enhances type 1 immune responses. However, their potential in solid tumors, including gastric cancer, is limited due to the immune suppressive TME. The prognostic value of NK cell infiltration like other lymphocyte infiltration has been shown in many cancers. However, to our knowledge, virtually no study directly investigated their role in EAC although their potential has gained attention in other adenocarcinomas like pancreatic cancer [107]. One study saw a favorable association of NK cells with survival in gastric adenocarcinoma [108]. Another study linked NK cell infiltration to prolonged survival in esophageal cancer but without clear distinction between ESCC and EAC [108].

### 3.9. Secretory Molecules

As broached in the previous paragraphs, the cytokine signaling pathways in the TME determine the tumor behavior. The initial IL-1β of the inflammation caused by chronic injury leads to the upregulation of IL-6 and IL-8 in the microenvironment by stromal cells, CAFs and immune cells [11]. IL-8 initially recruits neutrophils that, in the long run, adapt to an immunosuppressive phenotype. IL-6 likely drives the activation of MDSC that in turn act tumorigenic by the induction of TAMs and T_reg_s via TGF-β, IL-6, IL-10 and IL-12.

Utilizing a tissue micro array of 72 EAC patient samples, Conroy et al. showed a higher expression of TGF-β in the stroma surrounding EAC than within the tumor but significantly higher expression of IL-10 within the tumor [107]. Accordingly, T_reg_s were enriched in the stroma and CD107a, a marker for cytotoxic degranulation, was reduced in the tumor [59]. Similarly, elevated IL-17 levels were detected. The presence of IL-17 has further implication, as this cytokine has a strong link to Th_17_ cells, which are indeed involved in EAC [109]. A study of Liu et al. showed that IL-17 signaling enhances EAC progression and invasiveness through NF-κB-mediated MMP2 and MMP9 activation [110]. NF-kB is a central transcription factor in the pro-inflammatory activation of the immune system. On the stromal side, it was shown that NF-κB-inhibition in myofibroblasts reduced inflammation in the microenvironment of BE, thereby attenuating the phenotype in the L2-IL-1β mouse model [33].

Molecules downstream of cytokine signaling relevant in EAC have been identified too. For instance, Th_17_ cells also produce IL-22, a member of the IL-10 family that signals via signal transducer and activator of transcription (STAT) 3 [111]. STAT3 is a key transcription factor for Th_17_ cells and shifts immune responses towards the pro-tumorigenic side. It interacts with TAMs by inducing the expression of IL-23, which in turn (together with TGF-β, IL-1β and IL-6) induces Th_17_ differentiation. STAT3 also inhibits the pro-inflammatory transcription factor NF-κB in TAMs and IL-12 expression in DCs [112]. In IL23R-exprssing T_reg_s, IL-23 activates STAT3, leading to the production of IL-10, which also signals through STAT3 [113,114,115]. Strikingly, in a recent publication, Bhat et al. demonstrated that AP-endonuclease 1 (APE1), a positive regulator of STAT3 activity, was transiently overexpressed in BE cells after bile acid exposure, but was constitutively overexpressed in EAC cells [116].

### 3.10. Microbiome

Finally, the microbiome has been increasingly appreciated for its role in cancer progression. In malignancies, especially cancers of the gastrointestinal tract, microorganisms are frequently displaced in the tissue [117]. Intriguingly, EAC incidence has risen with the advent of antibiotics and the decline in *Helicobacter pylori* infection rates, suggesting a potential role of the microbiome in disease manifestation and progression at the esophagogastric junction [118]. Mechanistically, BE progression is associated with the infiltration of CD11b+Gr1+ myeloid cells [11] and it was shown that these cells also respond to bacterial lipopolysaccharide, a component of the outer membrane of Gram-negative bacteria [119]. The esophageal microbiome of BE and EAC patients is furthermore characterized by a general increase in Gram-negative bacteria [120]. We investigated the fecal microbiome driven by our recent findings in the L2-IL1β mouse model of BE, where high-fat diet led to dysplasia independent of obesity by changing the gut microbiome and consequently the inflammatory microenvironment [49]. In humans, changes in the compositions of the major commensals of the phyla Firmicutes, Proteobacteria, Bacteroidetes, Actinobacteria and Fusobacteria were described [121]. An increase in gram negative bacteria, such as *Fusobacterium nucleatum,* has been postulated, which may trigger NF-kB signaling in BE through TLR4 activation [122]. Considering that different microbes thrive at varying oxygen- and pH-levels as well as present antimicrobial peptides, a translocation during progression from BE to EAC appears plausible. Therefore, elucidating whether microbes influence disease progression might have prognostic value and could potentially be therapeutically targeted.

### 3.11. Search Strategy

The literature searches were conducted between September 2018 and March 2022 in the database of PubMed. Searching terms included esophageal adenocarcinoma, Barrett’s esophagus, tumor microenvironment, carcinogenesis, and immunology. Sources were preferred that discriminated esophageal adenocarcinoma from squamous cell carcinoma.

## 4. The TME as an Opportunity for Cancer Prevention

Tumor diseases are the second most common cause of death worldwide after cardiovascular diseases. The development of tumors is a complex process in which several genetic and non-genetic factors play a role. Although there is enormous potential in primary prevention, most cancers cannot be avoided. Therefore, secondary prevention through early detection and early treatment is important. The goal is to detect cancer as early as possible to improve treatment success and survival. Ideally, screening strategies should be clinically effective at the individual level and cost-effective at the population level. The carcinogenesis of EAC provides a good example to analyze the potential of molecular secondary prevention and to test its utility. Patients with gastrointestinal tumor diseases (colon, stomach, esophagus) already receive regular screening endoscopy. Therefore, material collection is already established and a concept of molecular prevention would be extendable to these diseases and, most importantly, practical and feasible in clinical applications. The detection of metaplastic or inflammatory precursor lesion is an important risk factor and it seems reasonable to improve the surveillance strategy by additional identification of biomarkers. This would allow a clearer prognosis in the future and thus extend the intervals of surveillance endoscopies, minimizing costs and the burden for the patient.

With a lifetime risk of merely 5% to progress, but an 11-fold relative risk of cancer, the major problem in clinical management of BE is the need to distinguish patients likely to progress from those that will not [5,123]. This challenge is only growing as new technologies improve the detection of BE in the population, which increases the number of patients we are managing. None of the currently known clinical and endoscopic criteria have a sufficient predictive power to identify BE progressors in a clinically useful manner, highlighting the interest to identify novel molecular biomarkers. We know very little about how cancers evolve and what fuels progression from normal to pre-malignant and malignant tissue. To improve cancer prevention, we need to understand the dynamics of epithelial transformation over space and time, and the selective pressures within the (micro) environment that shape these evolutionary trajectories.

Especially in patients with GERD and/or BE, surveillance is a key recommendation, for which substantial resources are expended. Guidelines on the clinical management of BE have been postulated by the American College of Gastroenterology (ACG) [124]. So far, surveillance of BE is done by esophagogastroduodenoscopy (EGD), including biopsies for pathological evaluation [125]. Patients with BE frequently present with symptomatic heartburn and seek medical attention at early stages of disease. This is useful for the detection of dysplasia or cancer but does not allow a true risk prediction for those patients who do not have dysplasia but could still progress to EAC. Given the high prevalence of GERD symptoms in the general population and the low prevalence of BE in these patients, a targeted screening for early detection and treatment on the one hand and a cost-effective approach on the other hand is warranted. Better prediction of neoplastic progression would facilitate to focus surveillance on patients with a high risk of malignant transformation. To explore individualized cancer screening strategies, data from specific and well-defined patient cohorts need to be used. This may allow individualized approaches for screening and surveillance in the future, while maximizing the clinical- and cost-effectiveness. The integration of identified TME risk factors into clinical assessment could help to identify more high-risk patients for cancer prevention and avoid overdiagnosis for low-risk patients.

The effectiveness of EGD has been questioned because the mortality remains high and the procedure can be costly cumulatively [126]. A novel method to monitor the state of BE is the use of the cytosponge, as reviewed by Iqbal et al. [126] The cytosponge is an encapsulated sponge that can be swallowed by the patient. In the stomach, the capsule of the sampling device dissolves, the sponge expands and is then pulled back up through the esophagus. Along the way it takes up cells from the luminal surfaces, that later can be analyzed. One advantage of this method sticking out is that nurses can perform it. In a clinical trial, the marker trefoil-factor 3 (TFF3) was used to identify BE in GERD patients and was compared to the standard EGD procedure. Patients who tested positive for TFF3 went on with endoscopic diagnosis. The cytosponge method showed an improved detection of BE [126]. The cytosponge TFF3 test appears to be a feasible, safe, and a generally acceptable test for outpatient use in patients with GERD and probably also obesity as a risk factor for EAC. The procedure leads to improved detection of BE patients, thereby allowing a more proactive prevention approach in identification with minimal invasive treatment of dysplasia and early EACs. It needs to be discussed whether TME analysis could be integrated into such a screening tool as well. In short, optimized prevention of tumor disease comprises a staggered investigation strategy with individually applied clinical diagnostics. This includes a combination of cytosponge (TTF3) with characterization of patients at risk (new molecular markers), as well as endoscopy in suspicious cases.

Carcinogenesis is accompanied by mutations, but unfortunately the mutational landscape is not a good predictive tool in EAC. Commonly mutated genes are present in both non-dysplastic BE and EAC, and the predictive value remains unclear [127]. The only exceptions to this are p53 and SMAD4 [23,127,128,129]. In EAC, a common mutational signature of T:A > G:C transversions in a CTT setting has been suggested to be associated with a mutation pattern caused by acid or bile exposure in the context of gastroesophageal reflux [94,95,99]. Unbiased stratification of EAC based on the mutational signature profiles also resulted in three subgroups that show different biological features and clinical characteristics [118]. It is likely that the exposure to different risk factors (e.g., bile reflux, *H. pylori* infection) in the upper gastrointestinal tract has an impact on these signature profiles and could be used for cancer prevention. Studies of heredity EAC causes focus mostly on familial aggregates. These studies are limited in sample sizes but were able to show clusters of BE or EAC [127,130]. However, the underpinning cause might be a genetic susceptibility to develop GERD [130].

As argued here, early detection of a tumorigenic niche would define a groundbreaking novel modality that potentially allows to define a population at risk to develop EAC. As one example, the receptor CXCR4 has previously been implicated for its diagnostic value as its expression on both tumor cells and immune cells was correlated with disease progression [131]. Early stage lesions are often not recognizable with conventional white-light endoscopy [132]. We observed a gradual increase of CXCR4^+^ neutrophils and T cells, which is in line with other reports about the role of these inflammatory leukocytes during tumorigenesis. In the L2-IL-1β mouse model, we tested fluorescence molecular endoscopy for such an inflammatory microenvironment marker, combined with white-light endoscopy, for the potential to serve as a “red-flag” technique for early EAC detection [133]. Indeed, the administration of CXCR4-targeted peptide conjugated to Sulfo-Cy5 dye (MK007) allowed early detection of dysplastic lesions [131,134,135]. The results from ex vivo studies further demonstrated a potential for clinical translation of fluorescence molecular imaging by enabling pre-clinical investigation of EAC biomarkers.

### TME as a Target for Chemoprevention

Chemoprevention using safe agents as a primary approach, or adjunctive to endoscopic therapy, is an attractive option to avoid neoplastic progression. Pharmacologic approaches that are well tolerated have the potential for wider adoption and to have a greater impact in reducing the number of patients that develop EAC. In the context of overall preventive public health, this approach would be largely non-invasive and would lower costs significantly by reducing the frequency of endoscopic surveillance. However, progress in the development of chemoprevention therapies for EAC has been severely restricted by the low progression rates and associated hurdles to initiate significant clinical trials.

Next to avoiding a reflux-inducing diet, proton pump inhibitors (PPI) have been shown to be beneficial [99]. PPIs mainly block acid production in parietal cells of the stomach. Consequently, intraesophageal pH values rise above physiological levels while cell proliferation decreases and differentiation in BE increases. A differentiated cell state is considered desirable, as opposed to de-differentiation, which is a cancer-associated cell characteristic. However, the clinical importance of such favorable effects on these surrogate markers is not clear. Although PPIs have an excellent safety profile, some researchers express concern about a possible increased cancer risk due to dysbiosis resulting from a higher pH [136]. Changes in the microbiome that occur during EAC development may play a role too, as elaborated on earlier. However, a definite study is currently missing. Other causes of dysbiosis, like the use of antibiotics and their contributions to cancer risk, are debated as well [137]. For instance, results from a UK study hint at an increased risk after repeated administration due to decrease of protective Streptococcus. However, the study fails to distinguish between ESCC and EAC, making it challenging to extrapolate insights [137,138].

Many studies have supported the potential of chemoprevention with cyclooxygenase (COX) inhibitors (coxib) for EAC, including aspirin, non-selective non-steroidal anti-inflammatory drugs (nsNSAIDs), and selective COX-2 inhibitors. Coxibs could exert their antitumor effect either by reducing the risk of BE or the risk of EAC progression in BE patients. Despite their early promise, coxibs are of no clinical value at this point. On the other hand, aspirin may be beneficial, and we have demonstrated in the mouse model that anti-inflammatory treatment with COX inhibitors or IL-1β antagonists (Anakinra©) can prevent carcinogenesis [139]. As another example, a randomized phase III study of low- or high-dose aspirin in combination with low- or high-dose PPI (esomeprazole) was conducted to chemo-prevent Barrett’s metaplasia (AspECT) [140]. The results show that a combination of high-dose PPI and low dose aspirin, significantly and safely improved outcomes in patients with BE, but without specificity for cancer development.

While this effect may be attributed to a broad reduction of inflammation, another class of drugs, statins, have more direct mechanisms that may prevent cancer development. Statins are a class of drugs prescribed to patients with elevated cholesterol levels. Statins reduce cholesterol by inhibiting HMG-CoA reductase of the mevalonate pathway, which is the rate-limiting step in cholesterol biosynthesis [141]. This has inhibitory effects on downstream metabolites generated by this pathway, including isoprenoids. This has several effects that decrease cell proliferation. For instance, without the post-translational modification of Ras (isoprenylation), cyclin-mediated cell cycle progression is reduced [141]. In a meta-analysis, statins were consistently associated with a reduced risk of progression to EAC by 41% [141]. A particular study showed that a combination of statin and acetylsalicylic acid increased the protective effect even further [142].

Conclusively, lifestyle choices pose a major risk factor to develop cancer. In EAC, BE is supposed to be the most important precursor lesion, as it increases the EAC risk 30-fold compared to the general population [123]. BE, in turn, is caused by chronic esophageal reflux disease (GERD). Therefore, a diet and behavior should be avoided that worsens reflux, such as smoking or a high fat diet and physical inactivity.

## 5. Research Perspectives

Overall survival of cancer patients has drastically improved over the past few decades thanks to a better understanding of the TME and cancer immunotherapies. However, the prognosis of EAC remains dismal. Accumulating molecular and clinical data may allow advancements in precision medicine for the treatment of EAC [143]. Hurdles like the heterogeneity of BE tissues dampen the development of effective prevention or treatment strategies. The pre-clinical L2-IL-1β mouse model offers the opportunity to image the immune infiltration more thoroughly than, e.g., resection of neoplastic regions in the patient. Moreover, thanks to the remarkable similarity to the human pathology, it might allow for a more comprehensive picture of the spatial and temporal distribution of immune cells in BE and EAC.

New advancements in gastro-esophageal cancer research have been fueled by the use of organoid systems derived from patient material, e.g., colorectal cancer [144]. This could help to reduce the use of animals. Although mouse models and organoid cultures are good tools to study EAC, results should be treated with caution while translating them to the clinic [16]. The published studies in the field coherently show the immune-suppressive phenotype on EAC. However, few studies provide answers to the spatial distribution of participating immune cells. Cao et al. discriminated the presence of macrophages in the tumor center or at the edge and found no difference in overall survival or nodal spread [101]. Based on sequencing data, MDSCs occur in EACs, but little research has been done on their role in this specific type of cancer [61]. The immune score, that has been proven successful in colorectal cancer evaluation, only factors in the presence of CD8^+^ (memory) T cells but neglects other important immune cells such as NK cells or macrophages [59]. The first successful attempts exist to establish a more comprehensive picture for SCC [145]. Additionally included markers were for T helper cells (CD4), Tregs (Foxp3), and myeloid cells (CD33), as well as inhibitory (PD-1/PD-L1, Tim-3, LAG-3) and stimulatory checkpoints (OX-40, ICOS), and a marker for suppressive polarization (IDO). Whether such a broad-ranging panel is practical in the clinical management, and which markers are useful in EAC, remains open and needs further understanding.

## 6. Concluding Remarks

Despite its rareness, EAC is an important malignancy due to the high prevalence of precursor lesions such as GERD and BE and its high mortality. Understanding the origin of this disease as an aberrant expansion of cardiac-derived gastric stem cells, and considering similar genomic and genetic alterations, allows us to group EAC with gastric cancer [14]. This has implications for the development of new prevention targets and future studies or clinical trials. Assuming BE and GERD as the dominant precursor lesions, disease progression follows a multistep process. Several pro- and anti-inflammatory processes are at play in the development of metaplasia or the progression to dysplasia and EAC. Knowledge on the role of specific immune cells and cytokines in the TME during that process are still fragmentary. The discussed tumor-promoting inflammation, avoidance of immune destruction, activation of invasion and metastasis, induction of angiogenesis and genomic instability are all “hallmarks of cancer” [143]. These hallmarks have been extended by Hanahan and Weinberg over time, e.g., by “polymorphic microbiomes”, which seems to be important for EAC too. Future studies will ideally confirm the discussed findings and put them together in a bigger picture. Such a comprehensive description of the TME in gastric junctional cancer and esophageal carcinogenesis would eventually make effective prevention possible.

## Figures and Tables

**Figure 1 cancers-14-02246-f001:**
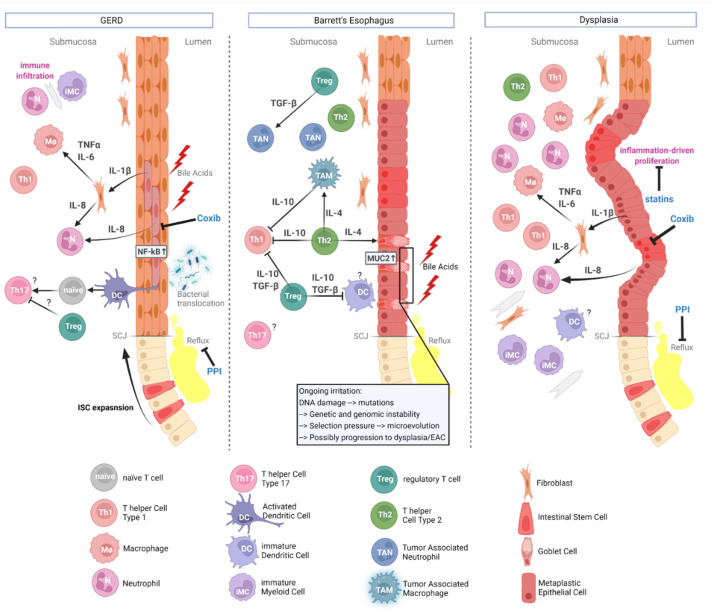
Immunologic networks in the progression of Esophagitis over Barrett’s esophagus to Dysplasia. Overall, the phenotype changes from type 1 domination in esophagitis (GERD) to type 2 in Barret’s Esophagus and back to type 1 in Dysplasia. Tregs may be initially protective through inhibition of tumorigenic Th_17_ cells. Th_2_ cells support metaplasia via IL-4, which induces goblet cell associated MUC2. Gray arrows indicate migration and question marks indicate unclear roles, as elaborated in the main text. Abbreviations: Coxib = Cyclooxygenase Inhibitor; IL = Interleukin; ISC = Intestinal Stem Cell; PPI = Proton Pump Inhibitor; SCJ = Squamo-Columnar Junction; TGF-β = Transforming Growth Factor β; TNFα = Tumor Necrosis Factor α. Generated with biorender.com.

**Figure 2 cancers-14-02246-f002:**
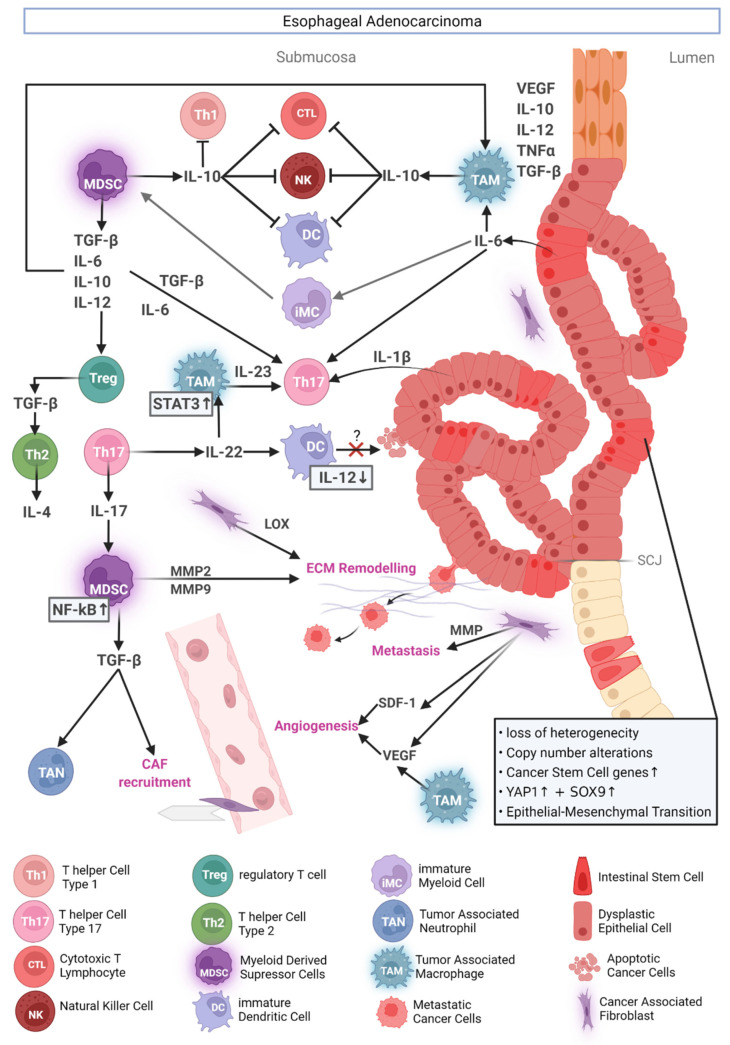
Tumor Microenvironment in Esophageal Adenocarcinoma. Redundant processes suppress central events in anti-tumor immunity, such as CTL and NK cell activity and associated Th1 activation, or DC maturation. Abbreviations: CAF = Cancer Associated Fibroblast; ECM = Extracellular Matrix; IL = Interleukin; LOX = Lysyl Oxidase; PPI = Proton Pump Inhibitor; SCJ = Squamo-Columnar Junction; SDF-1 = Stromal Cell Derived Factor 1; TGF-β = Transforming Growth Factor β; SOX9 = SRY-Box Transcription Factor 9; TNFα = Tumor Necrosis Factor α; VEGF = Vascular Endothelial Growth Factor; MMP = Matrix Metalloproteinase; and YAP1 = yes-associated protein 1. Generated with biorender.com.

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
