# Peer review of "Impact of the Tumor Microenvironment for Esophageal Tumor Development—An Opportunity for Prevention?"

_cancers, 2022, doi:10.3390/cancers14092246_

Round 1

Reviewer 1 Report

The authors of this manuscript are trying to give a comprehensive review about tumor microenvironment in esophageal adenocarcinoma. The Introduction section is already clear and provide sufficient background to conduct this review. All of the Discussion (from pathophysiology until potential target therapy) is also written in comprehensive manner. The Figures provided by the authors also make the understanding of the manuscript become easier. I think this manuscript is already good enough and can be accepted in its current form.

Author Response

Dear Reviewer 1,

As written in the attached letter, we want to thank you very much for the positive feedback.

Best wishes,

Martin Borgmann

Reviewer 2 Report

REference for that most EAC pt do not have evidence of BE at time of diagnosis (line 75-77)

Very comprehensive review. Excellent diagrams and cartoons. Clear. No concerns

Author Response

Dear Reviewer 2,

We want to thank you very much for the positive feedback. As detailed in the attached letter, we applied your suggested issue. 

Best wishes,

Martin Borgmann

Reviewer 3 Report

This manuscript is a nicely reviewed EAC tumor microenvironment, cancer management, and prevention. 

There are only a few minor concerns as below:

  1. The authors should provide the search strategy to help make this review more comprehensive.
  2. In line 42, please note that smoking is one of the significant risk factors for progression from BE to EAC, according to a publication in 2011. Please see the statement from a population-based study in 2011. "In this population-based study, which sought to investigate the association between lifestyle exposures among BE patients and their subsequent risk of esophageal or gastric cardia adenocarcinomas or esophageal HGD, tobacco smoking emerged as the strongest risk factor for progression." Tobacco Smoking Increases the Risk of High-Grade Dysplasia and Cancer Among Patients With Barrett's Esophagus. Helen G. Coleman, gastro.2011.10.034. So maybe the authors could add this statement to the manuscript. 
  3. In line 45, "EAC poses a major global health burden because it has a relatively high mortality rate of ≥ 80 %, ranking as the 6th deadliest malignancy in 2015 [6]."—Authors could update this statement using the data from the publication below: Global Cancer Statistics 2020: GLOBOCAN Estimates of Incidence and Mortality Worldwide for 36 Cancers in 185 Countries, CA CANCER J CLIN 2021;71:209–249
  4. In line 37, the authors wrote: "In Esophageal Adenocarcinoma (EAC), the tumor progressively evolves in an inflammatory process from the precursor lesion Barrett's Esophagus (BE), which is primarily caused by chronic reflux (gastroesophageal reflux disease, GERD)." In line 76, the authors wrote: "However, most EAC patients do not have evidence of BE at the diagnosis, calling this current dogma into question." I am confused by these two statements. The first one says EAC evolves in BE, while the second says most EAC patients do not have BE. Are they against each other? Do we have more published data to explain it further?
  5. In line 231, ISC has been defined in line 85. So the authors do not need to use the full name to describe ISC again. 
  6. Please correct "tumour" to "tumor" in the manuscript for consistency. 

Author Response

Dear Reviewer 3,

We want to thank you very much for the positive feedback. As detailed in the attached letter, we applied your suggested issues.

Best wishes,

Martin Borgmann
